# Trait Mapping of Phenolic Acids in an Interspecific (*Vaccinium corymbosum* var. *caesariense* × *V. darrowii*) Diploid Blueberry Population

**DOI:** 10.3390/plants12061346

**Published:** 2023-03-16

**Authors:** Ira A. Herniter, Yurah Kim, Yifei Wang, Joshua S. Havill, Jennifer Johnson-Cicalese, Gary J. Muehlbauer, Massimo Iorizzo, Nicholi Vorsa

**Affiliations:** 1Department of Plant Biology, Rutgers University, New Brunswick, NJ 08901, USA; 2Department of Agronomy and Plant Genetics, University of Minnesota, St. Paul, MN 55108, USA; 3Phillip E. Marucci Center for Blueberry and Cranberry Research and Extension, 125a Lake Oswego Road, Chatsworth, NJ 08019, USA; 4Department of Horticultural Science, North Carolina State University, Raleigh, NC 27695, USA; 5Plants for Human Health Institute, North Carolina State University, Kannapolis, NC 28081, USA

**Keywords:** blueberry, trait mapping, chlorogenic acids, candidate genes, fruit quality, nutrition

## Abstract

Blueberries (*Vaccinium* sect. *Cyanococcus*) are a dietary source of phenolic acids, including chlorogenic acid (CGA) and related compounds such as acetylated caffeoylquinic acid (ACQA) and caffeoylarbutin (CA). These compounds are known to be potent antioxidants with potential health benefits. While the chemistry of these compounds has been extensively studied, the genetic analysis has lagged behind. Understanding the genetic basis for traits with potential health implications may be of great use in plant breeding. By characterizing genetic variation related to fruit chemistry, breeders can make more efficient use of plant diversity to develop new cultivars with higher concentrations of these potentially beneficial compounds. Using a large interspecific F1 population, developed from a cross between the temperate *V. corymbosum* var. *ceasariense* and the subtropical *V. darrowii*, with 1025 individuals genotyped using genotype-by-sequencing methods, of which 289 were phenotyped for phenolic acid content, with data collected across 2019 and 2020, we have identified loci associated with phenolic acid content. Loci for the compounds clustered on the proximal arm of Vc02, suggesting that a single gene or several closely associated genes are responsible for the biosynthesis of all four tested compounds. Within this region are multiple gene models similar to hydroxycinnamoyl CoA shikimate/quinate hydroxycinnamoyltransferase (HCT) and UDP glucose:cinnamate glucosyl transferase (UGCT), genes known to be involved in the CGA biosynthesis pathway. Additional loci on Vc07 and Vc12 were associated with caffeoylarbutin content, suggesting a more complicated biosynthesis of that compound.

## 1. Introduction

The species commonly referred to as blueberry (*Vaccinium* sect. *Cyanococcus*) are perennial flowering plants native to North America. In the last thirty years, blueberry production has increased by over 600%, to 858,886 tonnes of blueberry harvested worldwide in 2020, with over 70% of that production occurring in the United States of America, Peru, and Canada [1]. This increase in production has occurred in the wake of the publication of a number of studies, beginning in the mid-1990s, examining the potential health benefits associated with blueberry consumption, especially in regard to antioxidant activity, for which blueberry has among the highest levels compared to other fruits and vegetables [2,3,4]. In addition to the use of the berries as fresh or processed foodstuff, the plants have also been used in traditional medicines [5]. Even today in the realm of natural medicine, blueberries are recommended for use in treating Type II diabetes [6].

Compounds with antioxidant activity, such as phenolic acids, anthocyanins, and anthocyanidins, have an important function in the plant. Plants regularly experience environmental stresses resulting in the production of reactive oxygen species (ROS), excessive levels of which disrupt normal metabolic function by damaging lipids, proteins, and nucleic acids, all of which negatively impact plant growth and development [7,8,9]. Compounds with antioxidant activity scavenge ROS, protecting normal metabolic function [10]. While the evidence for effect of antioxidant compounds obtained through food on human health is slim, the possibility of efficaciousness has encouraged substantial research on the topic.

One class of naturally occurring compounds known to have significant antioxidant capacity are phenolic acids, compounds containing a phenol moiety highly suited for trapping free radicals [10]. Chlorogenic acids are a family of polyphenol esters formed between *trans*-cinnamic acids and quinic acids [11]. Chlorogenic acids are one of the most well-studied families of polyphenols, due to their abundance in plant-based food and drinks [12]. They are widespread in plants and can be found in nearly all plant species [13,14,15,16].

One important subgroup of chlorogenic acids is the caffeoylquinic acids, which consist of esterifications of caffeic acid (Figure 1A), of which there are several isomeric forms. The most abundant of these isomers present in plants is 5-*O*-caffeoylquinic acid (5-CQA) [12], and it is the most widely studied due to its commercial availability [14]. Generally, the term “chlorogenic acid” (CGA) refers to 5-CQA [12]. It should be noted, however, that the nomenclature of the isomers 5-CQA (Figure 1B) and 3-CQA (3-*O*-cafffeoylquinic acid) (Figure 1C) can create some confusion. In 1976, the International Union of Pure and Applied Chemistry (IUPAC) reversed the order of the numbering of atoms on the quinic acid ring [17]. Consequently, the previously identified 3-CQA was renamed as 5-CQA [17], and the current 3-CQA refers to neochlorogenic acid in accordance with the new numbering system [12,16]. However, due to the name change, there is still considerable confusion in the literature, and many papers do not specify, when they use the term “chlorogenic acid”, whether they mean to refer to 5-CQA or 3-CQA [16]. This paper follows Clifford et al. [16] and uses the current IUPAC numbering, with CGA as 5-CQA.

Chlorogenic acids are the major hydroxycinnamic acids present in blueberries [18], likely accounting for a large proportion of their antioxidant activity [19,20], with CGA being the major component [21]. CGA constitutes 10–16% of total acids in the blueberry fruit [22,23,24]. It is present in concentrations of 98–208 mg per 100 g FW (fresh weight) in *V. corymbosum* cultivars [21,24,25]. Other phenolic acids such as caffeic acid, p-coumaric acid, and ferulic acid are present in concentrations under 1% [21,22].

CGA is one of the most abundant beneficial polyphenols in the human diet and is well known as a nutritional antioxidant in plant-based foods [26,27]. Dietary consumption of CGA is associated with the prevention of certain oxidative and degenerative, age-related diseases [28,29,30]. Compelling evidence indicates that dietary CGA can promote a wide range of pharmacological effects and biological activities in various tissues and organs [12]. Numerous studies have demonstrated the antioxidant activities of CGA, which include inhibiting the formation or scavenging of ROS [31]. CGA is also negatively correlated with the risk of various harmful conditions, such as oxidative and inflammatory stresses [32], type 2 diabetes mellitus [33,34], cardiovascular disease [35], neurodegenerative disease [36], and cancer [37].

Compounds closely related to CGA are the acetylated caffeoylquinic acids (ACQA, C_18_H_20_O_10_), which have previously been identified in blueberry [38]. These compounds have not been well characterized and the configurations of the compounds are not known, but the acetylation is likely to be on the quinic acid moiety [38]. Figure 1D shows a potential chemical structure for ACQA proposed by Jaiswal et al. [38]. Note that the regiochemistry of the acetyl group in Figure 1D is an arbitrarily selected example; the compound identified in the present analysis may not be the 4-acetyl caffeoylquinic acid isomer.

While CGA has been studied for its potential for improving human health, similar efforts have yet to be made to understand the potential health impacts of consumption of ACQA1 and ACQA2. Further research is required to understand the potential benefits of the compounds, as well as their bioavailability when consumed in food or drink.

Another related compound to CGA is caffeoylarbutin (CA). As with CGA, CA is an ester of caffeic acid, though with an arbutin group instead of quinic acid (Figure 1E). CA has been previously identified in the leaves of blueberry [39], as well as in the leaves of other *Vaccinium* species, such as lingonberry (*V. vitis-idaea*) [40,41], *V. dunalianum* [42,43], and bilberry (*V. myrtillus*) [39,41].

Other minor and specialty crops have seen great advances in the availability of genetic resources over the past decade. However, blueberry has lagged behind, with only a few published genome sequences [44,45,46] and a limited number of genotyped mapping populations [46,47,48,49], all of which were constructed through crosses between tetraploid cultivars of *V. corymbosum*. The development of mapping populations in blueberry is complicated by long generation time and partial self-sterility but is of exceeding importance to crop improvement efforts, as the identification of genetic markers and trait loci can increase the speed at which new varieties can be developed, and aid in understanding the mechanisms of compound biosynthesis.

The blueberry (*Cyanoccocus*) section of *Vaccinium* is highly diverse, including species of differing ploidy levels and adapted to different environments. The highly interfertile nature of the section offers great opportunities for geneticists and plant breeders alike to identify potentially valuable variations which could be used to develop new blueberry cultivars with improved nutritional value and increased resilience in the face of climate change. The most commercially important blueberries are tetraploid (4n = 48) highbush plants (*V. corymbosum*). Accordingly, most of the available evidence on the beneficial health effects of blueberry deals with compounds identified in tetraploids. However, many wild species exist throughout North America [50], naturally occurring as diploids, tetraploids, and hexaploids [20]. Among these are numerous diploid blueberry species which offer diverse germplasm [51], though most have limited analyses of their genetics. Two species likely to be of use in this effort are *V. darrowii* and *V. corymbosum*. *V. darrowii* is a subtropical lowbush diploid blueberry native to the American southeast which has been used as a source of variation in breeding programs [52,53], while *V. corymbosum* is a temperate highbush blueberry native to the mid-Atlantic region of the United States. The varying climates to which these two species are adapted, as well as genetic drift over evolutionary time, have resulted in wide differentiation on many traits including its fruit chemistry which is quite distinct from that of the highbush species [32,36]. This differentiation is exploited in this study by using hybrid populations of these two species for trait mapping. By identifying quantitative trait loci (QTL) in a population containing *V. darrowii* ancestry, greater insight can be obtained into the genetic architecture of blueberry.

A cost-effective method to produce genetic information for newly developed populations is genotyping-by-sequencing (GBS) as it simultaneously performs single nucleotide polymorphism (SNP) discovery and genotyping, eliminating a step required by other methods [54]. Using the newly available genome sequences, we can create highly accurate maps based on the physical ordering of the observed SNP markers. To the authors’ knowledge, phenolic acid content in blueberry has only been genetically mapped once previously, by Mengist et al. [48], who identified a QTL on Vc02 for CGA in an F_1_ tetraploid *V. corymbosum* population. While CA has been identified in a number of different species, as yet no QTL for CA has been identified. The ACQA compounds have not been fully characterized, and as yet have not been identified outside of blueberry, let alone been mapped.

In this paper, we present the first genotyped diploid interspecific mapping population derived from crosses between *V. darrowii* and *V. corymbosum* var. *caesariense* (a diploid variety of *V. corymbosum*), two divergent species. This large population, recently developed at the Marucci Blueberry and Cranberry Research Center in Chatsworth, NJ, segregates for many traits of interest to breeders, including fruit chemistry. We present here mapping of the genetic control of phenolic acid content in blueberry fruit.

## 2. Results

### 2.1. Qualitative and Quantitative Analysis of Phenolics

Figure 2 illustrates a selected HPLC chromatogram of blueberry phenolic compounds. Table 1 summarizes the retention times and MS spectra of the studied blueberry phenolic compounds. The exact isomer of caffeoylquinic acid could not be determined using the mass data. However, based on the literature, the peak identified was assumed to be CGA (5-CQA). Similarly, two of the peaks were identified as isomers of ACQA. To clarify which one is being referred to, these compounds have been labeled as ACQA1 and ACQA2 based on the order in which they were eluted.

Table 2 summarizes the mean and standard deviation values of all phenolic compounds among the genotypes. There were significant observed differences between average concentrations for all phenolic compounds in fruit among two genotypes of *V. corymbosum* var. *caesariense* (OPB-8 and OPB-15), two genotypes of *V. darrowii* (NJ88-12-41 and NJ88-14-03), and two interspecific hybrids (BNJ05-218-9 and BNJ05-237-8) (Figure 3). ACQA1 and ACQA2 concentrations were higher in the *V. corymbosum* var. *caesariense* blueberries, while CGA and CA concentrations were higher in *V. darrowii* blueberries, with intermediate levels in the hybrid blueberries (Appendix A). In the BNJ16-4 population, the mean CGA concentration was 0.25 ± 0.15 mg/g FW in 2019 and 0.23 ± 0.17 mg/g FW in 2020. The mean ACQA1 concentration was 0.14 ± 0.07 mg/g FW in 2019 and 0.13 ± 0.07 mg/g FW in 2020. The mean ACQA2 concentration was 0.11 ± 0.04 mg/g FW in 2019 and 0.09 ± 0.04 mg/g FW in 2020. The mean CA concentration was 0.05 ± 0.02 mg/g FW in 2019 and 0.05 ± 0.02 mg/g FW in 2020. Results for parents can also be found in Table 2. Graphs showing the range of concentrations found in the population and the parents can be found in Figure 3.

### 2.2. Trait Variation and Correlation

In the parents and grandparent plants there was no observed harvest year effect. The Kruskal–Wallis rank sum test did not identify significant differences between 2019 and 2020 in compound concentrations in the parents and grandparents considered together (Table 3). Despite this OPB-15 did show great variation, with observed phenolic compound concentrations reduced by about half. In the BNJ16-4 population, harvest year effects were observed for all the tested compounds except ACQA1.

In the parents and grandparents, each tested compound showed statistically significant correlations between itself and the other tested compounds (Table 4). ACQA1 and ACQA2 showed a strong correlation in both 2019 (0.97, *p* < 0.001) and 2020 (0.98, *p* < 0.001). CGA and CA also showed a strong correlation in 2019 (0.87, *p* < 0.001) and 2020 (0.85, *p* < 0.001). In addition, CGA and CA both showed moderate negative correlation with ACQA1 and ACQA2. In the BNJ16-4 population, ACQA1 and ACQA2 showed a strong correlation both in 2019 (0.89, *p* < 0.001) and 2020 (0.98, *p* < 0.001). CGA and CA showed a weak correlation in 2019 (0.26, *p* < 0.001) and a moderate correlation in 2020 (0.37, *p* < 0.001).

### 2.3. Map Information

The output VCF from the “population” function on STACKS produced a VCF with 47,594 markers, of which 47,199 were located on chromosome assemblies. After filtering to remove monomorphic markers in the parents (BNJ05-237-8 and BNJ05-218-9), or where one parent had a missing call, or where the minor allele frequency (MAF) > 0.05, 34,445 markers remained. Following processing using a custom R script to turn the data into the JoinMap format, 24,134 markers remained. Within JoinMap, the non-segregating markers were removed, leaving 17,633 markers. We further separated markers physical chromosomes, where markers on each chromosome were filtered to remove markers with >=0.95 similarity and significantly distorted markers (<1 × 10^−4^), leaving a total of 9952 markers for formation of linkage groups. The physical marker order was used to set the marker order and the Kosambi algorithm was used for map construction. During map construction, 7862 markers were removed for lack of information provided or were otherwise not mapped, leaving 2090 markers in the final map.

Of the 2090 markers in the final map, 771 markers were heterozygous in both parents, while 640 markers were heterozygous only in BNJ05-237-8, and 679 markers were heterozygous in only BNJ05-218-9. The total length of the map was 1591.6 cM, with the chromosomes ranging in length from 113.9 cM (Vc07) to 152.6 cM (Vc02), with an average length of 132.6 cM. Markers were generally well distributed across the genome, with an average of 0.8 cM between markers on both male and female parent-derived markers, with the exceptions of a 14.7 cM gap on Vc10, and gaps greater than 8 cM on Vc01, Vc06, Vc08, Vc11, and Vc12 (Appendix A). The distribution of markers and marker types across the genome is shown in Figure 4.

Graphs showing the relationship between physical and genetic maps of each chromosome can be found in Appendix A. Collinearity between the physical and genetic maps was very strong, with the Spearman’ correlation rho value being >0.999 for each chromosome. Polynomial formulae describing the instantaneous recombination rate across each chromosome were calculated and can be found in Appendix A. Most chromosomes displayed the characteristic “S” shape, with the ends of the chromosome demonstrating higher recombination rates than the centromeric regions. In most of the chromosomes, regions of lower recombination were located in or near the physical center of the chromosome, with the exception of Vc06, in which the region of reduced recombination was shifted towards the proximal end of the chromosome.

### 2.4. Trait Mapping and Heritability

Following phenotypic characterization and calculation of BLUPs, QTL were mapped using BLUPs and individual year data in R using the package *qtl* [55]. Peaks were identified for each tested compound (Table 5, Figure 5 and Appendix A). The identified peaks for CGA, ACQA1, and ACQA2 BLUPs clustered together on Vc02. The peaks for ACQA1 and ACQA2 BLUPs were identical, with a peak at 8.8 Mb with LOD scores of 26.8 and 23.6, and explaining 48.7% and 44.4% of the observed variation, respectively. The peak for CGA BLUPs was identified on Vc02 at 7.8 Mb with a LOD score of 17.7 and explaining 35.6% of the observed variation. Peaks for CA BLUPs were identified on Vc07 and Vc12, with LOD scores of 5.6 and 7.0, and explaining 12.9% and 16.0% of the observed variation, respectively. Mapping with phenotype data from individual years generally identified similar regions to the BLUPs, though sometimes with the peaks shifted up or downstream. One notable difference was that the 2020 CA data were associated with a peak on Vc02 with a LOD score of 5.15 explaining 9.7% of observed variation. The identified region on Vc02 overlaps with the region identified for CGA by Mengist et al. [48]. Complete marker-trait association LOD score data can be found in Appendix A. Broad sense heritability, calculated using the BLUPs, for the tested traits was strong: 0.894 for CGA, 0.945 for ACQA1, 0.948 for ACQA2, and 0.891 for CA.

### 2.5. Candidate Gene Identification

The minimal significant region on Vc02 (CGA, ACQA1, ACQA2), determined by where the mapped BLUP values overlapped with LOD scores above the threshold determined by permutation testing, stretched from 252,772 to 24,241,116 bp (23,988,344 bp). This region contains 1074 gene models. The region identified on Vc07 (CA) stretched from 16,286,998 to 29,752,448 bp (13,465,450 bp) and contained 474 gene models. The region identified on Vc12 (CA) stretched from 21,658,456 to 36,900,682 bp (15,242,226 bp) and contained 704 gene models. Within the significant region on Vc02, nine gene models had annotations matching known CGA biosynthesis pathway genes (Table 6). Seven had hits on hydroxycinnamoyl CoA shikimate/quinate hydroxycinnamoyltransferase (HCT), one from tea (*Camellia sinensis*) and six from coffee (*Coffea arabica*), while two had hits on UDP glucose:cinnamate glucosyl transferase (UGCT) from poplar (*Populus* spp.). Additionally, within the significant region on Vc02 were 15 gene models annotated as MYBs, a transcription factor family which has been associated with the phenylpropanoid biosynthesis pathway. The complete list of candidate gene models can be found in Appendix A.

## 3. Discussion

Phenolic acids are known to be potent antioxidants [10] with implications for human health. Breeding programs can make use of the marker-trait association presented in this study to develop marker-assisted breeding programs, potentially increasing the speed of the selection process and leading to the development of improved varieties with increased antioxidant potential.

The phenylpropanoid biosynthesis pathways have been extensively studied, with the pathways well understood [16,56,57,58], including both the enzymes involved, such as various transferases and hydroxylases and potential transcriptional regulators, such as those in the R2R3 MYB transcription family. Some potential pathways for the biosynthesis of ACQA1, ACQA2, and CA can be found in Figure 6.

Using the BLUPs, CGA, ACQA1, and ACQA2 had QTL located on the proximal arm of Vc02. Mapping for CA using BLUPs did not identify a peak on Vc02, though mapping with 2020 data did identify a QTL close to the others, and the 2019 data showed elevated correlation between CA and the ACQA compounds but one which did not rise to the level of significance. The overlapping QTL on Vc02 suggest that a single gene or several genes located in that region are involved in the biosynthesis of all four compounds. Notably, seven genes in the significant region on Vc02 had BLAST hits on HCT from tea (*Camellia sinensis*) and coffee (*Coffea arabica*). Previous studies have shown that HCT performs the final step in the main CGA biosynthesis pathway, producing caffeoyl-CoA, the penultimate compound in the path [16,56]. While this is the main pathway, others have been proposed [56]. Of relevance is the alternate pathway wherein cinnamic acid is bonded with glucose by UGCT. Two gene models in the significant region have BLAST hits on UGCT identified from poplar (*Populus* spp.). Unfortunately, while Mengist et al. [46] mapped CGA content and produced transcriptomes in tetraploid blueberry, they did not make a comparison between genotypes with varying levels of CGA and as yet, there is no searchable database with expression data in different tissues (J. L. Humann, personal communication).

R2R3 MYBs have been associated with the phenylpropanoid biosynthesis pathway [58,59]. Within the significant region identified on Vc02 are fifteen gene models annotated as MYB-encoding (Appendix A), offering additional avenues of potential exploration. Future studies should focus on the potential activity of such transcription factors in the biosynthesis of CGA, ACQA1, ACQA2, and CA in *Vaccinium*, perhaps identifying markers which can be used for breeding purposes.

Considerably less is known regarding the biosynthesis of other caffeoylquinic acid isomers. Our current understanding is that other caffeoylquinic acid isomers are derived from CGA, but there is little data on the enzymes that would be involved in such conversions [16]. In addition to the QTL identified on Vc02, CA also mapped to regions on Vc07 and Vc12, whereas the other compounds did not. This suggests that while CA biosynthesis is related to biosynthesis of the remaining compounds, it is more complicated. Further research is required to elucidate the biosynthesis pathways of the studied phenolic acids.

Comparing the results of trait mapping presented here with the previous findings of Mengist et al. [46] showcases the mapping power of the interspecific population. While Mengist et al. [46] used a population of 196 individuals and found QTL for CGA with LOD scores of 5.9–6.9, the present study used a comparable number of individuals and identified QTL for CGA with LOD scores of ~16.0. This difference is likely partially explained by the larger difference between the parents in the present study. While the parents in Mengist et al. [46] population, the cultivars Jewel and Draper-44392, differed by ~25mg/100g FW, BNJ05-237-8 and BNJ05-218-9 differed by ~110mg/100g FW. It is unlikely, however, that this would explain the 10-magnitude difference in probabilities. It is more likely that the diploid nature of the BNJ16-4 population allows for greater clarity. Amadeu et al. [60] demonstrates the increased accuracy of trait mapping in diploid as compared to tetraploid populations, suggesting that trait mapping efforts should be focused on diploid populations when possible. The BNJ16-4 population shows segregation for a wide range of traits, beyond those presented here. Future studies should make use of the population to map those traits with a similar degree of confidence.

In addition to its interspecific nature, the population used in this study offers advantages over the tetraploid populations used in other studies. This primarily stems from the diploid nature of the population, which greatly simplifies the genetic analysis. With just two copies of the genome, constructing maps is far simpler, as well as determining inheritance, dosage effects, and performance of QTL mapping [60]. Additionally, due to the relatively common failure of diploid male *Vaccinium* gametes to complete meiosis, ~13.5% of pollen grains have unreduced gametes [61], meaning that pollen from a diploid plant can be directly crossed with a tetraploid plant to bring the desired trait into the breeding population. Further, diploid blueberry can be induced to undergo polyploidization using oryzalin or colchicine [62]. Consequently, the bulk of breeding could potentially occur at the diploid level and the findings of mapping studies done in diploid plants, such as the present one, could be directly applied to the breeding program.

The nature of the map conformed to expectations from the literature. The centromeric regions, as defined by a region with large physical distances and minimal genetic distances, and hence a slope near zero, was generally found at or near the center of the physical map (Appendix A). This accords with the karyotype data from Hall and Galleta [63], who reported that centromeres in *Vaccinium* were, when observed, median to submedian. The one exception was Vc06, where the centromere was shifted toward the proximal arm of the chromosome.

The present study identifies a strong QTL for the tested phenolic acids, and its overlap with the QTL region identified by Mengist et al. [46] lends credence to our finding. However, the lack of independent populations in which markers could be tested means that any genetic selection, such as in a breeding program, would require using an individual or parent from the BNJ16-4 population as the donor. Further analysis on phenolic acid content in independent populations, such as biparental, multiparental, or germplasm collections, is required to develop widely applicable markers which could be used in marker-assisted selection.

The observed near 1:1 correlation between ACQA1 and ACQA2, in conjunction with both traits showing highly significant associations with the same region on Vc02, suggests the possibility that biosynthesis of both compounds is carried out by a single enzyme, and which of the two compounds is produced is random. The minor positive correlation between CGA and CA observed in the BNJ16-4 population (0.26 in 2019 and 0.37 in 2020, Table 4) and the minor negative correlation between CGA and the ACQA compounds (Table 4) suggest a relationship in the biosynthetic pathways of these compounds.

Phenolic content in OPB-15 varied dramatically between 2019 and 2020, with nearly double the concentration or more of each compound observed in 2019 as in 2020 (Table 2). This pattern was not observed in any of the other parents or grandparents. Previous studies have found that blueberries collected later in the season show increased levels of phenols, anthocyanins, and antioxidants [3,64,65,66]. Indeed, harvesting began and ended later in 2019 than in 2020. Another possible cause is elevated UV radiation in 2019 compared to 2020, which may have affected the phenolic content, though UV radiation was not measured in the greenhouse in which the plants were grown. Previous studies in carrot [67] and blueberry [68] have indicated that CGA content increases in correlation with increased UV radiation. It should also be noted that OPB-15 had low fruit yields in 2020, perhaps contributing or relating to the observed difference in CGA content. Furthermore, it is worth noting that the BNJ16-4 population also showed a general drop in phenolic compound concentrations in 2020 compared to 2019, with the exception of ACQA1, though not to the same extent as observed in OPB-15 (Table 2). This general decrease was significant, as evident in the results from the Kruskal–Wallis test (Table 3). Environmental variation could be involved in the inconsistent identification of a peak on Vc02 for CA. Further investigation on the possible environmental factors affecting phenolic content in blueberry is required, such as additional years of data or trials in different locations.

Similarly, CA is thought to have potentially beneficial effects for human health. CA levels are high in the leaves of bearberry and *V. dunalium*, and a tisane is made from the leaves of both in traditional medicine [39,69]. Arbutin and its derivatives, such as CA, are used as skin-whitening agents, due to their activity in inhibiting melanogenesis by inhibiting tyrosinase [39,70]; indeed, it has been shown to inhibit melanogenesis in zebrafish [71]. Arbutin has also been shown to have anticancer properties, likely due to its high level of antioxidant activity [72,73]. CA appears to be bioavailable, being present in urine following consumption [39]. This paper demonstrates the presence of CA in blueberry and suggests possible breeding avenues to increase CA content, with potential health benefits for consumers.

Each of the tested phenolic compounds contains a caffeoyl moiety (Figure 1A) as a component, suggesting that the significantly associated region encodes a gene responsible for the esterification of caffeic acid with various R groups. Further investigation could be directed at identifying other esters of caffeic acid in blueberry and mapping them to see if they co-locate.

Cultivated blueberry consists mostly of tetraploid *V. corymbosum*, while the population tested here was developed from crosses between diploids, *V. corymbosum* var. *caesariense* and *V. darrowii* plants. The observed difference in the grandparents, where OPB-8 and OPB-15 (*V. corymbosum* var. *caesariense*) showed elevated ACQA1 and ACQA2 concentrations and NJ88-14-03 and NJ88-12-41 (*V. darrowii*) showed elevated CGA and CA concentrations, could represent different strategies by the species to combat oxidative stress. In addition, CGA is known to be bioavailable in humans [14,74,75] while to the authors’ knowledge the bioavailability of ACQA1 and ACQA2 has not been tested. While further studies are required to determine the bioavailability of the ACQA compounds, should they be less bioavailable in humans or show lower antioxidant activity, breeding to increase CGA levels in blueberry cultivars could improve nutritional value of the berries. Wang et al. [51] compared various wild diploid species along with tetraploid cultivars of *V. corymbosum*, showing that the chemical composition of the tetraploids was distinct from that of the tested diploid species for anthocyanin and flavanol glycoside content. This could indicate that introgression from a variety collection, such as the grandparents of the BNJ16-4 population, would be of use in introducing variation to breeding programs.

## 4. Materials and methods

### 4.1. Plant Material

#### Population Development and Maintenance

A large F_1_ population derived from crosses between *V. corymbosum* var. *caesariense* and *V. darrowii* was used for this research. Two wild *V. corymbosum* var. *caesariense* plants, OPB-15 and OPB-8, were collected from a native population in Burlington County, NJ, near the Phillip E. Marucci Center for Research and Extension (39.71° N, 74.51° W). Two wild *V. darrowii* plants, NJ88-12-41 and NJ88-14-03, were collected from native populations in Liberty County, Florida (30.24° N, 85.01° W) and along Route 98S in the Saint Joseph Bay area of Florida (29.78° N, 85.28° W), respectively. Crosses between these plants were made in 2005, NJ88-14-03 x OPB-15, and OPB-8 x NJ88-12-41. The resulting hybrid plants were BNJ05-218-9 and BNJ05-237-8, respectively.

Reciprocal crosses of the F_1_ plants were made in 2016. Where BNJ05-218-9 was the female parent, plants were designated with the prefix “BNJ16-4”. Where BNJ05-237-8 was the female parent, plants were designated with the prefix “BNJ16-5”. The resulting F_1_ population consists of 1025 full-sib individuals, 949 BNJ16-4′s and 76 BNJ16-5′s. A scheme representing the pedigree of the population is presented in Appendix A.

Plants were maintained in pots in a greenhouse located at the Philip E. Marucci Blueberry and Cranberry Center for Research and Extension in Chatsworth, New Jersey, USA (39.72° N, 74.51° W). The plants went through the normal cycle of seasonal growth and winter dormancy (greenhouse maintained in a “cold” state, allowing for winter chilling at minimum 0–4 °C). Bumblebees (Koppert Biological Systems, Howell, MI, USA) were brought into the greenhouse in late spring/early summer during flowering for open pollination for fruit set.

### 4.2. Fruit Collection

A subset of genotypes was selected arbitrarily to be included in the phenotyping as it was not feasible to analyze every genotype. Berry samples were collected from 185 genotypes in both 2019 and 2020, 45 just in 2019, and 48 just in 2020, as not every plant produced fruit in both years of the study. Samples were also collected from the parents and F_1_ plants in both 2019 and 2020. Fully ripe, i.e., blue, fruit samples were harvested from each plant at 7–14-day intervals over the fruiting period. In 2019 fruit was harvested from the parents beginning on 9 April through 10 June, and from the BNJ16-4 subpopulation beginning on 4 June through 16 August, with a peak around 24 June. In 2020 fruit was harvested from the parents beginning on 1 May through 24 June, and from the 16-4 population beginning on 7 May through 2 July, with a peak around 4 June. Berry samples were placed in polyethylene bags and kept chilled until weight measurements were taken. After weighing, samples were stored at −80 °C until analysis. Information on the number of berries collected from each plant in both 2019 and 2020 can be found in Appendix A.

### 4.3. Genotyping and Map Construction

#### DNA Extraction and GBS

Young leaf samples from each genotype (BNJ16-4 and BNJ16-5 progeny, their parents, and grandparents) were collected in spring 2020 and DNA was extracted using a modified cetyltrimethylammonium bromide (CTAB) solution following Daverdin et al. (2017). DNA quantification was performed using a Qubit 3 fluorometer and the Qubit dsDNA BR assay kit (Invitrogen, Waltham, MA, USA).

Genotyping-by-sequencing (GBS) libraries were developed using a protocol derived from [76]. In brief, 200 ng of DNA was double digested using the restriction enzymes MspI and PstI-HF (New England Biolabs, Ipswich, MA, USA) at 37 °C for 2 h. A common adaptor and a unique barcode adaptor for each accession were ligated to the digested genomic DNA. Following ligation, the solutions were cleaned using 0.7 volumes of Axyprep Mag PCR Clean-Up magnetic beads (Axygen, Union City, CA, USA). The cleaned fragments were then amplified using Taq 5X Master Mix (New England Biolabs) and PCR primers with specific sequences to allow Flowcell binding and Illumina sequencing. After amplification, DNA from each accession was quantified again using the Qubit 3 fluorometer, then diluted to 5 ng/ul. Several pools were made using the diluted DNA from 112 samples with different barcodes. These pools were then then cleaned again using the Axyprep magnetic beads and quantified to ensure a concentration greater than 5 ng/ul. Pools of purified and barcoded genomic DNA were sequenced by Genewiz (South Plainfield, NJ, USA) in a 2 × 150 bp configuration on an Illumina Hiseq to produce paired end reads.

### 4.4. Demultiplexing, Sequence Alignment, and Variant Calling

GBS sequencing data were processed using STACKS v2 [77]. The *process_radtags* function was used to filter for quality, demultiplex, and trim the raw reads. The following options were used: -c to clean the data, removing any reads with uncalled bases; -q to remove low quality reads (phred ≥ 10); -r to rescue the barcodes and RAD-tags. Demultiplexed reads were then aligned to the reference genome of W85 [48], a wild diploid *V. corymbosum* var. *caesariense* accession collected in Ocean County, NJ, using SAMtools [78]. The obtained BAM files were processed in STACKS using the *ref_map.pl* command for SNP calling. To facilitate trait mapping, the outputs were formatted into VCF files using the *populations* command in STACKS with the following options: “--min-samples-per-pop 0.90” so that only SNPs present in at least 90% of the population would be retained; “--min-maf 0.2” so that only SNPs with a minor allele frequency >20% would be retained; “--ordered-export” to order the markers by physical location.

### 4.5. Map Construction

Map construction was performed using JoinMap v5 [79]. The VCF output file from STACKS was converted to JoinMap format in R. Prior to map construction, markers for which both parents were monomorphic or where either parent had missing data were filtered out. We examined progeny genotypes for patterns of coherent segregation and the presence of impossible genotypes, which were converted to missing data when aberrant genotypes were below a 5% threshold, otherwise loci above this threshold were removed from downstream analysis. A final step included the removal of markers with greater than 10% missing data in the progeny. The remaining markers were imported into JoinMap, with markers from each chromosome imported independently. In JoinMap, highly distorted (*p*-value ≤ 1 × 10^−5^) and highly similar (≥0.95) markers were removed. Maps were constructed using regression mapping with the Kosambi mapping function and with a fixed marker order based on the physical order within the reference genome. Default settings were used for ordering of markers within a linkage group, which include linkages with a recombination frequency <0.4000, a LOD of >1.00, a “goodness-of-fit” threshold of 5.00, followed by rippling after every added locus. The genotype data are available in Appendix A, and the map information is available in Appendix A.

### 4.6. Instantaneous Recombination Rate

A polynomial curve fitting the cM position as a function of physical location was generated in R version 4.1.3 (The R Foundation for Statistical Computing) for each of the 12 chromosomes. The linear model *lm* from the base *stats* package was used to determine the linear regression for polynomials up to the 25th degree. Then the output polynomials were examined to find the lowest degree polynomial with an R^2^ value greater than 0.998. The recombination rate (first derivative) of this polynomial was calculated along the length of the chromosome using the *deriv* function, also from the base *stats* package.

### 4.7. Phenolic Acid Extraction and Quantification

#### Chemicals and Reagents

All solvents, including water, acetonitrile, methanol, and acetone, were purchased from EMD Millipore (Billercia, MA, USA) and were of HPLC grade. Acetic acid was purchased from Avantor Performance Materials (Center Valley, PA, USA), and formic acid was purchased from Mallinckrodt baker (Phillipsburg, NJ, USA). A commercial standard of chlorogenic acid (CGA) was purchased from Sigma-Aldrich, Inc. (St. Louis, MO, USA).

### 4.8. Extraction of Blueberry Phenolic Compounds

For chlorogenic acid quantification, depending on sample availability, 2–8 g of fruits were weighed, average berry weight (AW) was recorded, and samples were ground with a Precellys Evolution homogenizer (Bertin Corp., Rockville, MD, USA) using 2.8 mm ceramic beads at 7200 rpm for 1.5 min. 80% aqueous acetone with 0.1% acetic acid (1:4 sample to solvent *w*/*v*) was added to suspend ground fruit, and samples were extracted overnight at 4 °C in a standard refrigerator. Liquid extracts were then centrifuged at 13,300 rpm for 2 min, and 1 mL aliquots of clear supernatant were taken. Aliquots were dried with a SpeedVac vacuum concentrator (Savant SPD2010-220, Thermo Scientific, Waltham, MA, USA) under no heat and re-dissolved in 500 uL of 100% methanol by sonication for 15 min. Samples were then centrifuged at 11,000 rpm for 5 min, and clear supernatants were analyzed with high-performance liquid chromatography (HPLC).

### 4.9. HPLC Apparatus and Conditions

Two HPLC systems were used for phenolic acid identification and quantification. The phenolic acids were analyzed in a Waters Alliance LC system composed of a Waters e2695 Separations Module and Waters 2998 PDA Detector (Waters Corp., Milford, MA, USA). A Gemini 150 × 4.6 mm 5 μm C18 110 Å LC column (Phenomenex, Torrance, CA, USA) was used for separation, and compounds were detected at 366 nm. The injection volume was 10 μL.

For compound identification of phenolic acids, the samples were analyzed using the method described in Wang et al. [51] with a Waters ACQUITY^®^ UPLC I-Class system coupled with a Waters Vion Ion Mobility Quadrupole Time of Flight (IMS QTof) mass spectrometer (MS) (Waters Corp., Milford, MA, USA). The same column, solvent system, and elution gradient as described by Wang et al. [51] were used with the system for compound identification. In addition, a 1:3 splitter was used to direct one-fourth of the flow (0.25 mL/min) into the MS. Compounds were identified by liquid chromatography tandem mass spectrometry (LC-MS-MS) based on accurate masses, retention times, and UV absorbance at 305 to 390 nm. All solvent systems and elution gradients are summarized in Table 7.

### 4.10. Compound Identification with MS Spectrometry of Samples

Ion-Mobility High-Resolution Mass Spectrometry data were acquired in high-definition MS^E^ mode, with the following parameters: ion source, ESI negative ion; analyzer type, sensitivity; source temperature, 100 °C; desolvation temperature, 400 °C; cone gas flow, 50 L/h; desolvation gas flow, 850 L/h; capillary voltage, 2.50 kV; low collision energy, 6.0 eV; high collision energy, 15.0–45.0 eV; mass range, 50–2000 *m*/*z*; scan rate, 0.25 s. Leucine encephalin (50 pg/mL, 10 μL/min) was used for lock mass correction at 0.25 min intervals. MS and ion mobility data were acquired and processed in UNIFI (Waters Corp., Milford, MA, USA).

### 4.11. Compound Characterization and Quantification

Phenolic acid characterization was carried out by comparing LC retention times, UV spectra and/or MS/MS data with standards (Table 1). For quantification of phenolic compounds, chromatograms were viewed at absorbance 366 nm and quantified as equivalents of their available standard, chlorogenic acid. The concentration of each compound was expressed in milligrams of its equivalent external standard per gram of fresh weight sample.

### 4.12. Phenolic Acid Data Analysis

Statistical analyses were performed using R version 4.1.1 (The R Foundation for Statistical Computing) and Microsoft Excel for Microsoft 365 MSO (New York, NY, USA). The *lme4* [80], *lmerTest* [81], and *emmeans* [82] packages were used to fit a linear regression model to the data, determine whether a significant difference exists between genotypes with Satterthwaite’s method, and run post-hoc analyses on differences between pairs of genotypes using Tukey multiple comparison tests with Kenward-Roger’s degree of freedom method. The *corrplot* [83] package was used to generate a Pearson’s correlation matrix among the phenolic concentrations. Correlations were classified as strong if r > |0.7|, moderate if |0.3 > r > 0.7|, or weak if r < |0.3|. Kruskal–Wallis H tests were used to evaluate harvest year and localization effects on individual compound concentrations in the parents and F_1_ plants. Excel was used to generate tables using mean and standard deviation values derived from the data. Excel was also used to generate frequency distribution histograms and tables using F_1_ data.

Best linear unbiased estimates (BLUPs) were calculated using the lme4 package in R. Only genotypes with phenotypic data from both 2019 and 2020 were used to calculate BLUPs, with harvest year considered as a fixed variable. Broad-sense heritability was calculated using the variance components as follows, modified from Mengist et al. [46]:(1)$$H^{2}=\frac{\partial_{g}^{2}}{\partial_{g}^{2}+\frac{\partial_{gy}^{2}}{y}+\frac{\partial_{e}^{2}}{s}}$$
where $\partial_{g}^{2}$, $\partial_{gy}^{2}$, and $\partial_{e}^{2}$, are the variance components of genotypes, genotype-by-environment interactions, and environment, respectively; *y* is the number of years the plants were phenotyped (2 years of the study) and *s* is the total number of samples in the data set.

### 4.13. Trait Mapping

Trait mapping was done using the R package *qtl* [55]. Data were imported using the “read.cross” function, treating the population as a four-way cross and incorporating information about marker phase, and the “jittermap” function was used to separate markers at the same cM location. The probability value of each SNP was determined with the function “calc.genoprob (data, step = 0).” Afterward, to map the QTL probabilities, both a standard interval mapping using the EM algorithm: “scanone (data)” and a Haley-Knott regression: “scanone (data, method = “hk”, n.cluster = 2)” were used. Both algorithms showed similar results. To test for significance, 1000 permutations were performed on the Haley–Knott regression: “scanone (data, method = “hk”, n.perm = 1000)”. Only SNPs with LOD scores greater than the significance threshold determined through the permuation test were considered significant. Trait mapping was performed on calculated BLUPs and individual year data. Phenotypic data used in this study are available in Appendix A. SNPs with LOD scores above the significance threshold were considered to be significant.

### 4.14. Candidate Gene Identification

The QTL mapping results for each trait and for each year were compared to identify the region containing overlap between significant regions as determined by the LOD scores. The list of annotated gene models from the W85 *(V. corymbosum* var. *caesariense)* sequence [48] was surveyed to identify lists of genes located in the overlapping significant regions as determined by the permutation test. The gene annotations on this list were then examined for similarity with genes known from the literature to be involved in the CGA biosynthesis pathway.

## 5. Conclusions

This study analyzed phenolic acid content and identified marker-trait associations in an interspecific population developed from crosses between *V. corymbosum* var *caesariense* and *V. darrowii*. The QTL identified explained 9.7–48.7% of the observed variation. We identified overlapping peaks on Vc02 for all the tested compounds, with additional peaks for CA on Vc07 and Vc12. This suggests that located within the significant region on Vc02 is a gene, or cluster of genes, which plays a major role in the phenolic acid biosynthesis pathway. This study demonstrates the applicability of interspecific populations to trait mapping. The identified QTL can be used in breeding programs to improve the nutritional value of newly developed cultivars.

## Figures and Tables

**Figure 1 plants-12-01346-f001:**
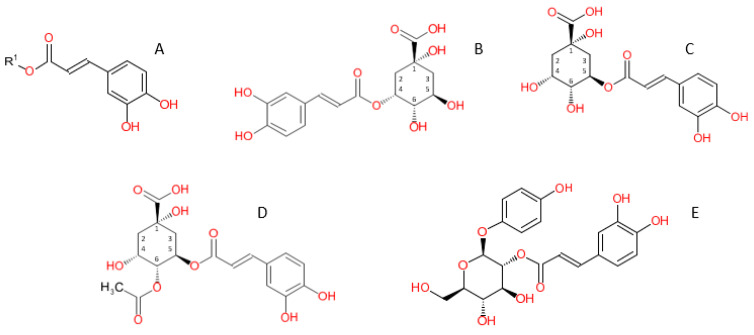
Chemical structures of esterification products of caffeic acid. (**A**) caffeic acid R = H, (**B**) 3-*O*-caffeoylquinic acid (neochlorogenic acid), (**C**) 5-*O*-caffeoylquinic acid (chlorogenic acid), (**D**) acetyl-caffeoylquinic acid (tentative), (**E**) 2-O-caffeoylarbutin.

**Figure 2 plants-12-01346-f002:**
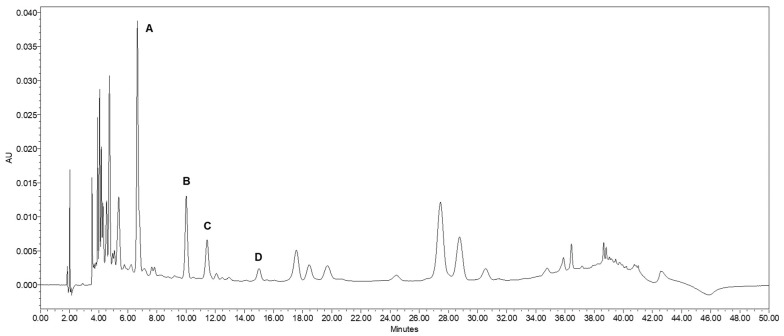
Representative PDA chromatogram of blueberry accession BNJ05-218-9 acquired at 366 nm. The characteristic peaks are labelled 1 through 4. (A) chlorogenic acid, (B) acetyl-caffeoylquinic isomer 1, (C) acetyl-caffeoylquinic isomer 2, (D) caffeoylarbutin.

**Figure 3 plants-12-01346-f003:**
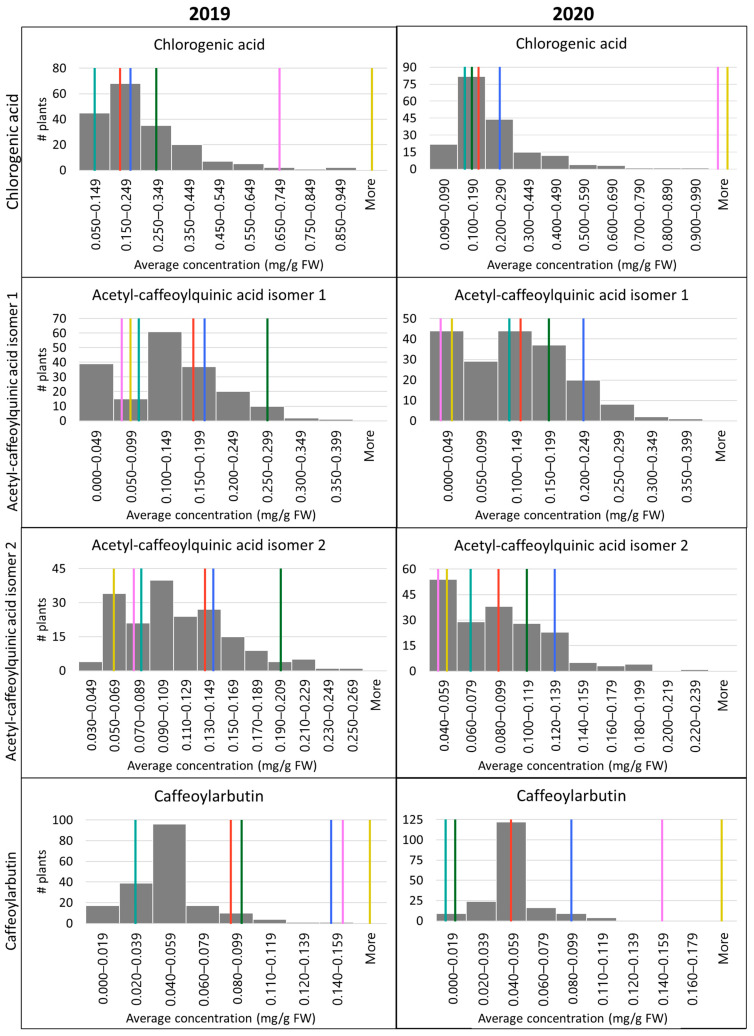
Range and frequency of concentrations of tested compounds in the BNJ16-4 population (mg/g fresh weight). Values from the grandparents and F_1_s are noted with vertical lines in the appropriate bins: OPB-8 (teal), OPB-15 (green), NJ88-14-03 (pink), NJ88-12-41 (yellow), BNJ05-218-9 (red) and BNJ05-237-8 (blue).

**Figure 4 plants-12-01346-f004:**
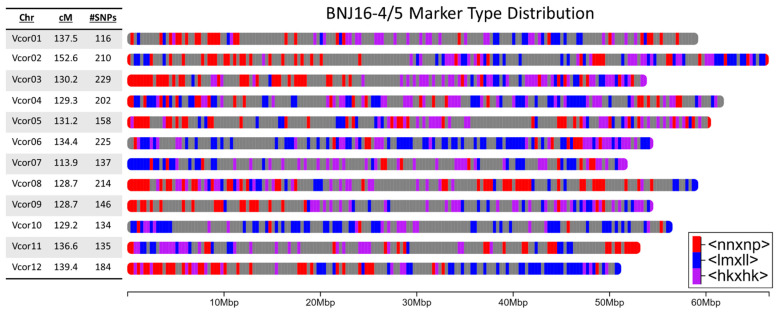
Distribution of markers in the genetic map across the physical chromosomes. <nnxnp> markers are those which are homozygous in BNJ05-237-8 and heterozygous in BNJ05-218-9, <lmxll> are the opposite, and <hkxhk> are heterozygous in both parents.

**Figure 5 plants-12-01346-f005:**
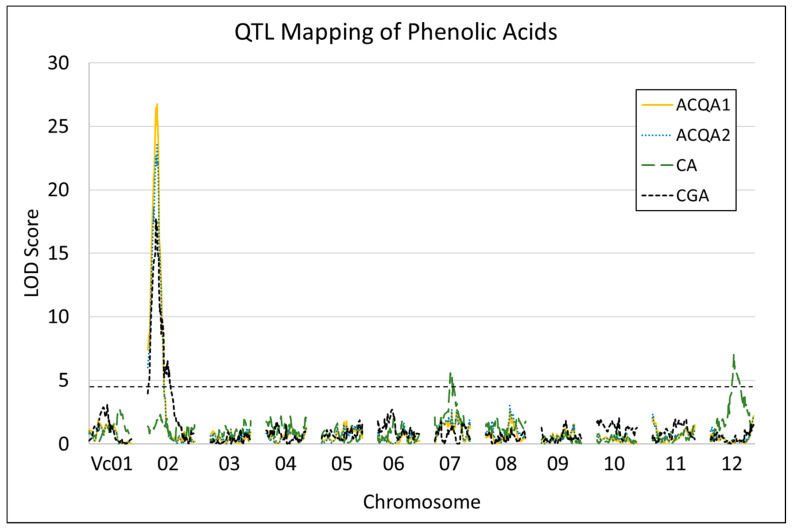
Trait mapping results for BLUPs of the phenolic acids. The horizontal dashed line indicates the significance cutoff (~4.5).

**Figure 6 plants-12-01346-f006:**
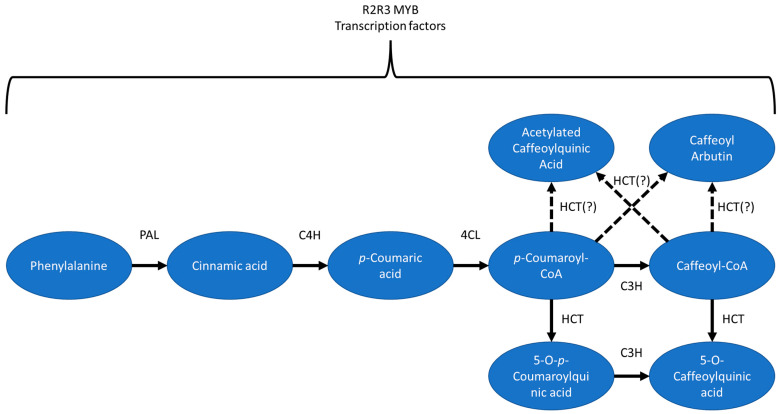
Some of phenolic acid biosynthesis pathways. Solid lines with arrows represent known relationships while dashed lines with arrows represent possible biosynthesis pathways based on the mapping data. Enzymes: phenylalanine ammonia lyase (PAL), cinnamate 4′-hydroxylase (C4H), 4-cinnamoyl CoA ligase (4CL), hydroxycinnamoyl CoA: quinate hydroxycinnamoyl transferase (HCT), and p-coumaroyl-3′-hydroxylase (C3H). Various R2R3 MYB transcription factors are believed to play regulatory roles in the various biosynthesis pathways.

**Table 1 plants-12-01346-t001:** Phenolic acids identified in blueberries. ESI-MS-MS: electrospray ionization-tandem mass spectrometry.

Peak Number	Retention Time (min)	[M-H]	Main Fragment Ions in ESI-MS-MS (*m*/*z*)	Formula	ppm	Tentative Peak Identity
1	6.657	353.0873	191.0556 (100)	C_16_H_18_O_9_	−1.3	Chlorogenic acid (5-*O*-caffeoylquinic acid, 5-CQA, CGA)
2	10.010	395.098	191.0556 (100), 233.0662 (75)	C_18_H_20_O_10_	−1.0	Acetyl-caffeoylquinic acid isomer 1 (ACQA1)
3	11.437	395.0979	233.0662 (100)	C_18_H_20_O_10_	−1.2	Acetyl-caffeoylquinic acid isomer 2 (ACQA2)
4	15.005	433.1139	161.0236 (100), 323.0763 (9)	C_21_H_22_O_10_	−0.3	Caffeoylarbutin (CA)

**Table 2 plants-12-01346-t002:** Mean, median, standard deviation, and range of phenolic acid concentrations (mg/g fresh weight) in blueberry fruit harvested in 2019 and 2020.

2019		*V. corymbosum* var. *caesariense*		Hybrids.		*V. darrowii*		F1
	**Genotype**	**OPB-8**	**OPB-15**	**BNJ05-237-8**	**BNJ05-218-9**	**NJ88-12-41**	**NJ88-14-03**	**BNJ16-4**
**CGA**	Range	0.04–0.21	0.23–0.32	0.21–0.28	0.14–0.42	0.47–1.76	0.18–1.27	0.06–0.90
	Mean	0.09	0.27	0.24	0.24	1.03	0.72	0.25
	Median	0.07	0.27	0.24	0.21	1.01	0.57	0.22
	SD	0.05	0.04	0.03	0.1	0.29	0.36	0.15
**ACQA1**	Range	0.04–0.23	0.24–0.33	0.18–0.23	0.13–0.28	0.03–0.15	0.02–0.12	0.02–0.40
	Mean	0.1	0.29	0.2	0.19	0.07	0.07	0.14
	Median	0.08	0.28	0.19	0.15	0.05	0.04	0.13
	SD	0.06	0.04	0.03	0.07	0.03	0.03	0.07
**ACQA2**	Range	0.04–0.18	0.17–0.23	0.11–0.18	0.09–0.21	0.03–0.11	0.02–0.13	0.04–0.25
	Mean	0.08	0.2	0.15	0.14	0.06	0.07	0.11
	Median	0.06	0.18	0.15	0.12	0.05	0.06	0.11
	SD	0.04	0.03	0.03	0.05	0.03	0.03	0.05
**CA**	Range	0–0.10	0.05–0.11	0.10–0.17	0.05–0.13	0.17–0.60	0.03–0.22	0–0.15
	Mean	0.03	0.09	0.14	0.09	0.39	0.15	0.05
	Median	0.03	0.1	0.14	0.1	0.39	0.15	0.04
	SD	0.03	0.02	0.03	0.03	0.11	0.06	0.02
**2020**	**Genotype**	**OPB-8**	**OPB-15**	**BNJ05-237-8**	**BNJ05-218-9**	**NJ88-12-41**	**NJ88-14-03**	**BNJ16-4**
**CGA**	Range	0.06–0.16	0.06–0.24	0.13–0.41	0.08–0.20	0.91–1.60	0.51–1.49	0.05–1.26
	Mean	0.1	0.13	0.26	0.15	1.22	1.08	0.23
	Median	0.1	0.11	0.29	0.17	1.19	1.13	0.18
	SD	0.04	0.06	0.08	0.04	0.23	0.28	0.17
**ACQA1**	Range	0.08–0.19	0.08–0.30	0.15–0.29	0.11–0.18	0.04–0.05	0.04–0.05	0.04–0.37
	Mean	0.12	0.17	0.21	0.14	0.05	0.04	0.13
	Median	0.11	0.17	0.23	0.13	0.05	0.04	0.12
	SD	0.04	0.08	0.04	0.03	0	0	0.07
**ACQA2**	Range	0.05–0.11	0.06–0.18	0.09–0.17	0.07–0.13	0.04–0.05	0.04–0.05	0.04–0.23
	Mean	0.07	0.11	0.12	0.09	0.05	0.04	0.09
	Median	0.07	0.1	0.12	0.09	0.05	0.04	0.08
	SD	0.02	0.04	0.03	0.02	0	0	0.04
**CA**	Range	0–0.04	0–0.04	0.07–0.11	0.05–0.06	0.21–0.42	0.11–0.18	0–0.17
	Mean	0.01	0.02	0.09	0.06	0.33	0.15	0.05
	Median	0	0	0.09	0.06	0.34	0.15	0.04
	SD	0.02	0.02	0.01	0	0.07	0.02	0.02

**Table 3 plants-12-01346-t003:** Kruskal–Wallis rank sum test with phenolic acid concentrations (mg/g fresh weight) in blueberry fruit by harvest year (alpha = 0.05) in the parents, grandparents, and BNJ16-4 population and genotype-by-environment (GxE) variation in the BNJ16-4 population.

	Parents and Grandparents			BNJ16-4 F_1_s			
Compound	χ^2^	df	*p*-Value	χ^2^	df	*p*-Value	GxE Variation
CGA	0.00	1	1	21.09	1	4.38 × 10^−6^	3.66 × 10^−3^
ACQA1	0.41	1	0.52	2.20	1	0.14	4.71 × 10^−4^
ACQA2	1.64	1	0.2	50.95	1	9.48 × 10^−13^	1.28 × 10^−4^
CA	0.92	1	0.34	8.82	1	0.00298	8.17 × 10^−5^

**Table 4 plants-12-01346-t004:** Correlational data for the Parents and Grandparents and the BNJ16-4 population between tested compounds. Italicized numbers indicate the *p*-values of the observed correlations. *p*-value < 0.05 was considered significant. Correlations were ranked as weak (|<0.3|), moderate (|0.3–0.7|), or strong (|>0.7|).

**Parents**									
**2019**	**CGA**	**ACQA1**	**ACQA2**	**CA**	**2020**	**CGA**	**ACQA1**	**ACQA2**	**CA**
**CGA**					**CGA**				
**ACQA1**	−0.36				**ACQA1**	−0.62			
	*(<0.001)*					*(<0.001)*			
**ACQA2**	−0.30	0.97			**ACQA2**	−0.55	0.98		
	*(0.003)*	*(<0.001)*				*(<0.001)*	*(<0.001)*		
**CA**	0.87	−0.35	−0.34		**CA**	0.85	−0.48	−0.42	
	*(<0.001)*	*(<0.001)*	*−0.001*			*(<0.001)*	*(<0.001)*	*(<0.001)*	
**BNJ16-4**									
**2019**	**CGA**	**ACQA1**	**ACQA2**	**CA**	**2020**	**CGA**	**ACQA1**	**ACQA2**	**CA**
**CGA**					**CGA**				
**ACQA1**	−0.11				**ACQA1**	−0.02			
	*0.027*					*0.672*			
**ACQA2**	−0.09	0.89			**ACQA2**	0.03	0.98		
	*0.064*	*(<0.001)*				*0.491*	*(<0.001)*		
**CA**	0.26	0.12	0.07		**CA**	0.37	0.03	0.03	
	*(<0.001)*	*0.014*	*0.12*			*(<0.001)*	*0.61*	*0.595*	

**Table 5 plants-12-01346-t005:** Significant QTL identified. The chromosome on which the QTL is located, the marker interval of significant SNPs (above the LOD cutoff), the range and length of the significant interval in both cM and bp, the peak SNP, its location in cM and bp, the LOD score of the peak SNP, the LOD score cutoff for significance, and the percent observed phenotypic variation explained by the QTL are shown.

Trait	Chr	Start Marker	End Marker	Range (cM)	Range (bp)	Length (cM)	Length (bp)	Peak SNP	Peak (bp)	Peak (cM)	Peak LOD	LOD Cutoff	% Phenotypic Variation
acqa1_2019	2	91154:71:-	126708:471:-	0–51.1	242,415–24,241,116	51.13	23,998,701	105300:86:+	8,829,280	29.8	29.93	4.53	45.1
acqa1_2020	2	91154:71:-	126708:471:-	1–51.1	242,415–24,241,116	51.13	23,998,701	103495:459:-	7,840,716	26.1	24.27	4.60	38.1
acqa1_BLUP	2	91154:71:-	126708:471:-	2–51.1	242,415–24,241,116	51.13	23,998,701	105300:86:+	8,829,280	29.8	26.78	4.38	48.7
acqa2_2020	2	91154:71:-	126708:471:-	3–51.1	242,415–24,241,116	51.13	23,998,701	103495:459:-	7,840,716	26.1	20.10	4.60	32.8
acqa2_BLUP	2	91154:71:-	126708:471:-	4–51.1	242,415–24,241,116	51.13	23,998,701	105300:86:+	8,829,280	29.8	23.57	4.49	44.4
acqa2_2019	2	91210:108:-	126708:471:-	2.4–51.1	252,772–24,241,116	48.77	23,988,344	105300:86:+	8,829,280	29.8	25.50	4.61	40.0
cga_BLUP	2	91210:108:-	149574:48:-	2.4–73.1	252,772–38,430,662	70.75	38,177,890	103495:459:-	7,840,716	26.1	17.65	4.55	35.6
cga_2019	2	92629:11:-	138543:529:+	3.6–57.0	1,060,018–32,042,845	53.32	30,982,827	108869:604:+	11,684,755	35.8	15.90	4.51	27.3
cga_2020	2	94530:8:+	152995:69:-	6.3–75.9	2,175,510–40,705,298	69.61	38,529,788	103495:459:-	7,840,716	26.1	16.00	4.45	27.0
ca_2020	2	108869:604:+	113794:161:+	35.8–40.4	11,684,755–14,961,103	4.55	3,276,348	113794:161:+	14,961,103	40.4	5.15	4.33	9.7
ca_2019	7	562033:52:+	586588:110:+	42.0–58.3	11,162,618–28,613,768	16.34	17,451,150	569836:55:+	16,286,998	50.1	6.42	4.42	12.1
ca_2020	7	569836:55:+	591183:501:-	50.1–62.8	16,286,998–32,079,646	12.70	15,792,648	569836:55:+	16,286,998	50.1	5.48	4.33	10.3
ca_BLUP	7	569836:55:+	588225:292:+	50.1–60.2	16,286,998–29,752,448	10.10	13,465,450	569836:55:+	16,286,998	50.1	5.55	4.48	12.9
ca_2019	12	984802:89:+	1014237:66:+	57.9–98.6	18,054,908–38,178,446	40.62	20,123,538	997344:51:-	26,910,209	75.6	7.93	4.42	14.7
ca_2020	12	990220:105:+	1020637:545:-	68.9–111.1	21,657,929–42,473,585	42.27	20,815,656	997276:67:-	26,881,646	75.3	7.06	4.33	13.0
ca_BLUP	12	990220:632:+	1012122:503:+	69.1–94.3	21,658,456–36,900,682	25.21	15,242,226	997344:51:-	26,910,209	75.6	7.02	4.48	16.0

**Table 6 plants-12-01346-t006:** Candidate genes on Vc02. Nine candidates with annotations related to the CGA biosynthesis pathway were located within the significant region on Vc02. The gene name, annotation, the species from which the annotation was derived, and physical location are shown.

Sequence Name	Description [Species]	Location (bp)
Vcev1_p0.Chr02.03043	UDP-glycosyltransferase 73B4-like [*Populus euphratica*]	2,761,603–2,764,485
Vcev1_p0.Chr02.03044	UDP-glycosyltransferase 73B4-like [*Populus euphratica*]	2,771,253–2,774,928
Vcev1_p0.Chr02.03666	hydroxycinnamoyl-CoA:shikimate/quinate hydroxycinnamoyltransferase [*Camellia sinensis*]	16,397,202–16,398,569
Vcev1_p0.Chr02.03737	hydroxycinnamoyl-CoA shikimate/quinate hydroxycinnamoyl transferase [*Coffea arabica*]	18,658,014–18,660,014
Vcev1_p0.Chr02.03738	hydroxycinnamoyl-CoA shikimate/quinate hydroxycinnamoyl transferase [*Coffea arabica*]	18,666,233–18,668,469
Vcev1_p0.Chr02.03741	hydroxycinnamoyl-CoA shikimate/quinate hydroxycinnamoyl transferase [*Coffea arabica*]	18,749,921–18,753,906
Vcev1_p0.Chr02.03744	hydroxycinnamoyl-CoA shikimate/quinate hydroxycinnamoyl transferase [*Coffea arabica*]	18,912,803–18,914,710
Vcev1_p0.Chr02.03861	hydroxycinnamoyl-CoA shikimate/quinate hydroxycinnamoyl transferase [*Coffea arabica*]	22,777,759–22,779,946
Vcev1_p0.Chr02.03864	hydroxycinnamoyl-CoA shikimate/quinate hydroxycinnamoyl transferase [*Coffea arabica*]	22,898,343–22,901,060

**Table 7 plants-12-01346-t007:** Solvent system and elution gradient for HPLC analysis of phenolic acids in blueberry. Time point indicates time since sample begins running.

Time Point (min)	Flow (mL/min)	0.1% Formic Acid in Water (% Solution)	0.1% Formic Acid in Acetonitrile (% Solution)
0	1	100	0
1	1	86	14
5	1	85	15
14	1	85	15
20	1	84.6	15.4
30	1	83	17
35	1	73	27
38	1	60	40
40	1	20	80
43	1	100	0
50	1	100	0
80	0	100	0

## Data Availability

The data presented in this study are available in the Appendix A.

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
