# Peer review of "Trait Mapping of Phenolic Acids in an Interspecific (Vaccinium corymbosum var. caesariense × V. darrowii) Diploid Blueberry Population"

_plants, 2023, doi:10.3390/plants12061346_

Round 1

Reviewer 1 Report

It is good study but need to improve the whole article. My suggestions are in attached pdf

Reviewer 2 Report

A manuscript entitled Trait mapping of phenolic acids in an interspecific (Vaccinium corymbosum var. caesariense x V. darrowii) diploid blueberry population aims to assess For the first genotyped diploid interspecific mapping population derived from crosses between V. darrowii and V. corymbosum var. caesariense (a diploid 145 variety of V. corymbosum), two divergent species. This large population recently developed at the Marucci Blueberry and Cranberry Research Center in Chatsworth, NJ, segregates for many traits of interest to breeders, including fruit chemistry. We present here 148 mapping of the genetic control of phenolic acid content in blueberry fruit.

The manuscript is well-written. Overall, the manuscript has provided good structure by good implies the importance of Understanding the genetic basis for Vaccinium corymbosum var. caesariense x V. darrowii traits use in plant breeding..

 Minor comments:

-          Please add some results in abstract

-          Resolution of photo is not good.

-          Please check typo and Italic name in all article.

Round 2

Reviewer 1 Report

article has  revised considering the comments.

Author Response

We would like to thank the reviewer for their comments and suggestions. The manuscript is greatly improved thanks to their input. Your time and effort are greatly appreciated.

Sincerely,

Ira A. Herniter